

# Disentangling the effect of host-genotype and environment on the microbiome of the coral *Acropora tenuis*

Bettina Glasl[1,2,3], Caitlin E. Smith[1,2,3], David G. Bourne[1,2,3] and Nicole S. Webster[1,3,4]

[1] Australian Institute of Marine Science, Townsville, QLD, Australia
[2] College of Science and Engineering, James Cook University, Townsville, QLD, Australia
[3] AIMS@JCU, Townsville, QLD, Australia
[4] Australian Centre for Ecogenomics, University of Queensland, Brisbane, QLD, Australia

Corresponding author
Bettina Glasl, b.glasl@aims.gov.au

## ABSTRACT

Genotype-specific contributions to the environmental tolerance and disease susceptibility of corals are widely accepted. Yet our understanding of how host genotype influences the composition and stability of the coral microbiome subjected to environmental fluctuations is limited. To gain insight into the community dynamics and environmental stability of microbiomes associated with distinct coral genotypes, we assessed the microbial community associated with *Acropora tenuis* under single and cumulative pressure experiments. Experimental treatments comprised either a single pulse of reduced salinity (minimum of 28 psu) or exposure to the cumulative pressures of reduced salinity (minimum of 28 psu), elevated seawater temperature (+2 °C), elevated $pCO_2$ (900 ppm), and the presence of macroalgae. Analysis of 16S rRNA gene amplicon sequence data revealed that *A. tenuis* microbiomes were highly host-genotype specific and maintained high compositional stability irrespective of experimental treatment. On average, 48% of the *A. tenuis* microbiome was dominated by *Endozoicomonas*. Amplicon sequence variants (ASVs) belonging to this genus were significantly different between host individuals. Although no signs of stress were evident in the coral holobiont and the vast majority of ASVs remained stable across treatments, a microbial indicator approach identified 26 ASVs belonging to Vibrionaceae, Rhodobacteraceae, Hahellaceae, Planctomycetes, Phylobacteriaceae, Flavobacteriaceae, and Cryomorphaceae that were significantly enriched in corals exposed to single and cumulative stressors. While several recent studies have highlighted the efficacy of microbial indicators as sensitive markers for environmental disturbance, the high host-genotype specificity of coral microbiomes may limit their utility and we therefore recommend meticulous control of host-genotype effects in coral microbiome research.

# INTRODUCTION

Corals contain abundant and diverse communities of microorganisms that together form a holobiont (*Rohwer et al., 2002*). The photoautotrophic dinoflagellate endosymbionts

of the family Symbiodiniaceae are by far the best studied symbiotic partners of reef-building corals. Symbiodiniaceae lineages vary between coral species (*Smith, Ketchum & Burt, 2017*) and even between host genotypes of conspecific corals (*Brener-Raffalli et al., 2018*). Fine-scale adaptations of the Symbiodiniaceae lineages can influence the environmental sensitivity of their hosts (*Baker, 2003*), as some Symbiodiniaceae lineages are more thermo-tolerant and hence infer higher bleaching tolerance to corals (*Rowan, 2004*). Corals also harbor diverse communities of bacteria, archaea, and viruses (*Bourne, Morrow & Webster, 2016*; *Hernandez-Agreda, Gates & Ainsworth, 2017*; *Thurber et al., 2017*). Excessive environmental stress resulting in coral bleaching, tissue necrosis, and mortality, is often accompanied by a shift in the microbiome (*Glasl, Herndl & Frade, 2016*; *Zaneveld, McMinds & Thurber, 2017*). While the importance of the microbiome to coral fitness is well appreciated (*Bourne, Morrow & Webster, 2016*; *Grottoli et al., 2018*; *Peixoto et al., 2017*; *Ziegler et al., 2017*), the microbiome's potential to expand the environmental tolerance of coral holobionts via microbial shuffling and switching is far less understood (*Webster & Reusch, 2017*). *Endozoicomonas*, a bacterial genus commonly associated with marine invertebrates, is considered a putative symbiont of corals as it can occur at high abundance in aggregates within the tissue (*Neave et al., 2016b*) and loss of *Endozoicomonas* is frequently seen in bleached or diseased corals (*Bayer et al., 2013*; *Glasl, Herndl & Frade, 2016*). Pangenome analysis of *Endozoicomonas* has revealed evidence for functional specificity between strains (*Neave et al., 2017*), hence fine-scale changes in the composition or relative abundance of different *Endozoicomonas* strains may contribute to variation in the environmental tolerance and disease susceptibility of conspecific corals.

A fundamental question in microbiome research is whether host intrinsic factors (e.g., genetics) or the environment are the main drivers of microbiome composition and stability (*Spor, Koren & Ley, 2011*; *Wullaert, Lamkanfi & McCoy, 2018*). The influence of host genetics and environmental factors on the community composition of a microbiome varies between host species and even between host compartments. For example, the rizhosphere microbiome of the perennial plant *Boechera stricta* are predominantly shaped by environmental factors, however, its leaf associated microbial community is largely controlled by host genetic factors (*Wagner et al., 2016*). Host-genotype specific factors also shape the gut microbiome of *Drosophila melanogaster*, a model system for animal-microbe interactions, and further mediate its nutritional phenotype (*Chaston et al., 2016*). While many coral microbiome studies have focused on the effect of environmental stress (e.g., elevated temperature, increased macroalgae abundance, anthropogenic pollution, and declining water quality (*Garren et al., 2009*; *Vega Thurber et al., 2009*; *Zaneveld et al., 2016*; *Zhang et al., 2015*)); the combined influence of host-genotype and environment on the microbial community composition remains largely unknown. This is a critical knowledge gap as microbiome-by-host genotype-by-environment interactions may have important implications for the resistance of corals to stress and disease. Considering the recent declines in coral reefs (*De'ath et al., 2012*; *Hoegh-Guldberg et al., 2007*; *Hughes et al., 2017*) and the key role microorganisms play in maintaining host health (*Bourne, Morrow & Webster, 2016*),

disentangling the effect of environment and host-genotype on a coral's microbiota is of utmost importance.

This study investigated the effect of host genotype-by-environment interactions on the microbiome of *Acropora tenuis*. The compositional variability of the *A. tenuis* microbiome associated with distinct host genotypes (individual coral colonies) was assessed with high taxonomic resolution based on amplicon sequence variants (ASV). The stability of the microbiome was further investigated by exposing corals to acute salinity fluctuations (ranging from 35 to 28 psu) under current (sea surface temperature of 27.5 °C and $pCO_2$ of 400 ppm) and future (sea surface temperature of 29.5 °C, $pCO_2$ of 900 ppm, and macroalgae) projected reef conditions. Stress treatments were designed to simulate environmental conditions that *A. tenuis* can experience in their natural environment. Both stress treatments (single and cumulative stress) consisted of a non-lethal low salinity pulse, mimicking freshwater influx into the reef as occurs after large rainfall events, often linked to cyclones that cross the Eastern Australian coastline and result in large riverine flows into the nearshore and mid-shelf reef areas of the Great Barrier Reef (*Jones & Berkelmans, 2014*; *VanWoesik, DeVantier & Glazebrook, 1995*).

## MATERIALS AND METHODS

### Coral colony collection and experimental design

Nine *A. tenuis* colonies were collected from Davies Reef (Great Barrier Reef, Australia) in March 2017 and transported to the National Sea Simulator at the Australian Institute of Marine Science (Townsville, QLD, Australia). Corals were fragmented into coral nubbins, glued onto aragonite plugs and kept at control temperature (27.5 °C) and light (150 mol photons $m^{-2}$ $s^{-1}$) conditions in indoor flow-through aquaria for 3 weeks to allow healing. Corals were collected under the permit G12/35236.1 granted by the Great Barrier Reef Marine Park Authority to the Australian Institute of Marine Science.

The experimental design consisted of three treatment conditions: (1) control, (2) single stress, and (3) cumulative stress treatment (Fig. 1). Nubbins of all nine *A. tenuis* genotypes (A–I) were exposed to all three treatment conditions to explore microbiome variation according to host genotype. Each experimental aquarium (three aquaria per treatment) held nubbins of three *A. tenuis* genotypes (four nubbins per genotype, total of 12 nubbins per aquarium). Coral nubbins were acclimated to experimental aquaria for 3 weeks during which corals in the cumulative stress treatment were gradually ramped to 29.5 °C and 900 ppm $pCO_2$ over a period of 12 days. Corals in the control and single stressor treatments were kept at stable temperature (27.5 °C) and ambient (400 ppm) $pCO_2$ conditions throughout the experiment.

Salinity was ramped down over 3 h to a minimum of 28 psu and oscillated between 28 and 30 psu in a 6 h rhythm to simulate natural fluctuations occurring on reefs (tidal influences). Temperature and $pCO2$ adjusted freshwater (0.2 μm filtered) was used to lower salinities prior to supplying the low saline seawater to the aquaria tanks. After 7 days of low salinity, the salinity was ramped up (3 h) to 35 psu. In the cumulative stress treatment, corals were additionally exposed to elevated temperature (29.5 °C), $pCO_2$

**Figure 1 Conceptual overview of the experimental design.** *Acropora tenuis* colonies (*n* = 9) were fragmented and coral nubbins of each host genotype (A–I) were exposed to three different treatment conditions (control, single stress, and cumulative stress) and sampled on a regular basis throughout the experiment (day 1, 10, 14, and 19). Image credit: Bettina Glasl.

(900 ppm), and macroalgae (*Sargassum* sp.), as predicted for the end of the 21st century (*IPCC, 2014*).

Samples were collected regularly throughout the experiment (see Fig. 1), including 24 h before the salinity pulse was induced (day 1) and at three time points (day 10, 14, and 19) after the low-salinity stress exposure. All nubbins were processed as follows: effective quantum yield was measured (pulse amplitude modulation (PAM) fluorometry), photographed, inspected for visual signs of stress (tissue lesions, bleaching, and necrosis), rinsed with 0.2 μm filter-sterilized seawater, snap frozen in liquid nitrogen and stored at −80 °C until further processing.

Coral nubbins were defrosted on ice before tissue was removed with an airgun in 1 × PBS (pH = 7.4), homogenized for 1 min at 12.5 rpm with a hand-held tissue homogenizer (Heidolph Silent Crusher M) and subsequently aliquoted for the quantification of Symbiodiniaceae cell density, chlorophyll *a*, protein concentration, and DNA extraction for amplicon-based sequencing of the 16S rRNA gene. Aliquots (500 μl) for Symbiodiniaceae cell counts were fixed with formaldehyde (final concentration 1.5%) and stored in the dark at room temperature. Aliquots for chlorophyll *a*, protein, and DNA extraction (1 ml each) were centrifuged for 10 min at 16,000×*g*, the supernatant was discarded and the remaining pellet was snap frozen in liquid nitrogen and stored at −80 °C until further processing. Coral nubbin surface area was assessed by a single paraffin wax dipping for 2 s followed by 5 min air-drying. The weight of each coral nubbin before and after dipping was quantified and the surface area was calculated against a standard curve.

## Physiology of Symbiodiniaceae and the coral holobiont

The effective quantum yield of the Symbiodiniaceae was measured using PAM fluorometry. Corals were light adapted (5 h) before measuring the response of the photosystem II effective quantum yield (ΔF/Fm′) with a Heinz Walz™ Imaging PAM as previously described (*Chakravarti, Beltran & Van Oppen, 2017*). Coral nubbins were exposed to a saturation pulse and the minimum and maximum fluorescence was recorded and effective quantum yield was calculated (see Eq. S1).

Symbiodiniaceae cell densities were manually counted under a stereomicroscope using formaldehyde fixed tissue samples (final *c* = 1.5%). Samples were briefly vortexed

and 9 µl of each sample was added to either side of two haemocytometers and the density of symbiont cells was quantitatively normalized to the tissue blastate and aliquot volume, and standardized to the nubbin's surface area.

Chlorophyll *a* was extracted and concentrations were measured using a spectrophotometric assay. Tissue pellets were defrosted on ice, centrifuged at $16,000 \times g$ for 10 min at 4 °C, and remaining supernatant was discarded. Pellets were re-suspended in 1 ml of 100% acetone and incubated in the dark for 24 h at 4 °C after which they were centrifuged at $16,000 \times g$ for 10 min and supernatant (200 µl) was pipetted into a 96-well plate in triplicate. Absorbance at 630 and 663 nm was measured using a BioTek™ microplate reader and chlorophyll *a* concentration was calculated (see Eq. S1), quantitatively normalized to the tissue blastate and aliquot volume, and standardized to the nubbin's surface area.

Total protein concentration was quantified using a Pierce™ BCA Protein Assay Kit (Thermo Scientific, Waltham, MA, USA) following the manufacturer's instruction. Absorbance was measured in triplicate for each sample at 562 nm in a BioTek™ Plate reader. Standard curves were calculated using a bovine serum albumin (BSA) solution, creating a working range between 20 and 2,000 µg ml$^{-1}$ and total protein was calculated against the BSA standard curve, quantitatively normalized to the tissue blastate and aliquot volume, and standardized to the surface area of each individual nubbin.

## DNA extraction, 16S rRNA gene sequencing and analysis

DNA of all coral samples was extracted using the DNeasy PowerBiofilm Kit (QIAGEN, Venlo, Netherlands) following the manufacturer's instructions. Blank extractions were included to control for kit contamination. Coral DNA extracts were stored at −80 °C until shipment on dry ice to Ramaciotti Centre (University of New South Wales, Australia) for sequencing. The V1–V3 region of the 16S rRNA gene was amplified using primers 27F (5′-AGAGTTTGATCMTGGCTCAG-3′; *Lane, 1991*) and 519R (5′-GWATTA CCGCGGCKGCTG-3′; *Turner et al., 1999*) and libraries were prepared with the Illumina TruSeq protocol, followed by Illumina MiSeq 2 × 300 bp sequencing (see Table S1).

Demultiplexed paired end reads were analyzed in QIIME2 (Version 2017.9.0; https://qiime2.org) as previously described by *Glasl et al. (2018)*. In brief, forward and reverse reads were truncated at their 3′ end at the 296 and 252 sequencing positions, respectively. Samples were checked for chimeras and grouped into features based on 100% sequence similarity, from here on referred to as ASV, using DADA2 (*Callahan et al., 2016*). Multiple de novo sequence alignments of the representative sequences were performed using MAFFT (*Katoh et al., 2002*). Non-conserved and highly gapped columns from the alignment were removed using default settings of the mask option in QIIME2. Unrooted and rooted trees were generated for phylogenetic diversity analysis using FastTree. For taxonomic assignment, a Naïve-Bayes classifier was trained on the SILVA v123 99% operational taxonomic units, where reference sequences only included the V1–V2 regions (27F/519R primer pair) of the 16S rRNA genes. The trained classifier was applied to the representative sequences to assign taxonomy. A total of 11,063,364 reads were retrieved from 100 sequenced samples and clustered into 4,624 ASVs (Table 1). Chloroplast and

**Table 1 Sequencing and sample overview.**

| Host-genotype | Total no. of samples | No. of sequences | Richness[a] | Evenness[a] | Shannon Index[a] |
|---|---|---|---|---|---|
| A | 12 | 54,352 (±18,259) | 71 (±64) | 0.63 (±0.05) | 2.53 (±0.48) |
| B | 12 | 31,702 (±19,058) | 51 (±44) | 0.66 (±0.14) | 2.49 (±0.86) |
| C | 12 | 26,421 (±26,065) | 108 (±86) | 0.73 (±0.11) | 3.23 (±0.65) |
| D | 12 | 59,543 (±28,560) | 101 (±102) | 0.64 (±0.07) | 2.74 (±0.80) |
| E | 12 | 27,348 (±24,386) | 100 (±110) | 0.69 (±0.10) | 2.97 (±0.81) |
| F | 12 | 36,097 (±21,293) | 108 (±103) | 0.73 (±0.08) | 3.18 (±0.84) |
| G | 4 | 55,460 (±35,822) | 126 (±74) | 0.75 (±0.07) | 3.46 (±0.74) |
| H | 12 | 44,101 (±19,488) | 92 (±63) | 0.65 (±0.14) | 2.81 (±0.64) |
| I | 12 | 51,998 (±23,968) | 109 (±73) | 0.63 (±0.08) | 2.82 (±0.65) |

**Note:**
[a] Diversity indices (average ± SD) for each host genotype are based on a non-rarefied ASV table from which chloroplast and mitochondria derived reads were removed.

Mitochondria derived sequence reads and singletons were removed from the dataset and the feature table was rarefied to an even sequencing depth of 3,506 sequencing reads, leading to the exclusion of four samples. Demultiplexed sequences and metadata are available from the NCBI Sequence Read Archives under accession number PRJNA492377.

## Statistical analysis

Statistical analysis was performed in R (*R Development Core Team, 2008*). Holobiont health metadata were $z$-score standardized and variation between treatments and host genotypes was evaluated using Analysis of Variance (ANOVA) and if applicable, variations were further assessed with a Tukey post hoc test. Multivariate statistical approaches including Multivariate Homogeneity of Group Dispersion ("vegan package"; *Oksanen et al., 2013*), Permutation Multivariate Analysis of Variance (PERMANOVA, "vegan package"; *Oksanen et al., 2013*), Non-metric multidimensional scaling ("phyloseq package"; *McMurdie & Holmes, 2013*) and distance based Redundancy Analysis (db-RDA "phyloseq package"; *McMurdie & Holmes, 2013*) were based on Bray–Curtis dissimilarities. Mantel statistics based on Pearson's product-moment correlation (mantel test, "vegan package" (*Oksanen et al., 2013*) were used to evaluate whether sample-to-sample dissimilarities in microbiome composition and physiological holobiont health parameters (protein concentration, chlorophyll *a* concentration, Symbiodiniaceae cell densities, and effective quantum yield) were correlated. Holobiont health parameters were $z$-score standardized and dissimilarity matrices were based on Bray–Curtis dissimilarities.

Alpha diversity measures including richness and Shannon diversity for the *Endozoicomonas* community were analyzed using the "phyloseq package" (*McMurdie & Holmes, 2013*). Variation in the total relative abundance of all *Endozoicomonas* ASVs per sample between treatments, over time and between host-genotypes was assessed using ANOVAs with arcsine-square-root transformed relative abundance data. The phylogenetic tree of the 11 most abundant *Endozoicomonas* ASVs was produced with phyloseq (*McMurdie & Holmes, 2013*) using the Newick rooted tree generated in QIIME2 (version 2017.9.0; https://qiime2.org).
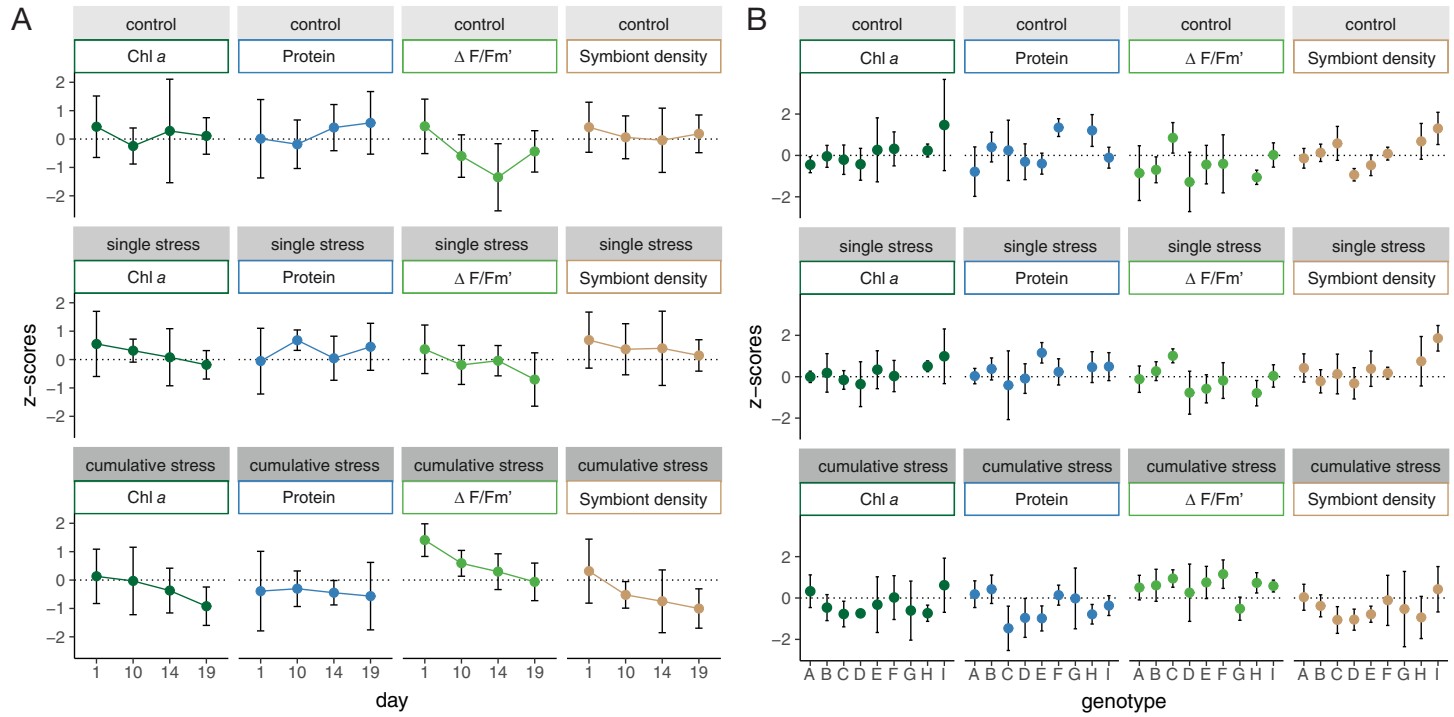

**Figure 2** **Physiological response of *Acropora tenuis* under control, single stress, and cumulative stress treatments.** Variations in the chlorophyll *a* (Chl *a*) concentration, protein concentration, effective quantum yield (ΔF/Fm′) and symbiont cell density (Symbiont density) of *A. tenuis* (A) over time (day 1, 10, 14, and 19) and (B) between individual host-genotypes (A–I). Physiological parameters are *z*-score standardized and error bars represent standard deviations.

Indicator value analysis (IndVal, "indispecies" package; *De Cáceres & Legendre, 2009*) was used to identify ASVs significantly associated with treatment groups (control, single stress, and cumulative stress) based on their occurrence and abundance distribution. Day 1 samples were excluded from the IndVal analysis to restrict the dataset to ASVs significantly associated with coral tissue after stress exposure (day 10, 14, and 19).

Graphs were created in R using ggplot2 (*Wickham, 2009*) and phyloseq packages (*McMurdie & Holmes, 2013*). Alluvial diagram was generated in RAWGraph (*Mauri et al., 2017*).

## RESULTS

### Coral holobiont physiological response

Corals showed no visual signs of stress (change in pigmentation, bleaching, tissue necrosis, and/or mortality) in any treatment. Chlorophyll *a* concentrations remained stable between treatments (one-way ANOVA with sampling time point as blocking factor, $F_{(2/94)} = 2.707$, $p = 0.072$), however, effective quantum yield (ΔF/Fm′; one-way ANOVA with sampling time point as blocking factor, $F_{(2/94)} = 15.52$, $p = 1.49 \times 10^{-6}$), symbiont cell densities (one-way ANOVA with sampling time point as blocking factor, $F_{(2/94)} = 8.83$, $p = 3.06 \times 10^{-4}$) and protein concentration (one-way ANOVA with sampling time point as blocking factor, $F_{(2/94)} = 5.563$, $p = 5.21 \times 10^{-3}$) varied significantly between treatments within sampling time points (Fig. 2A). Coral nubbins in

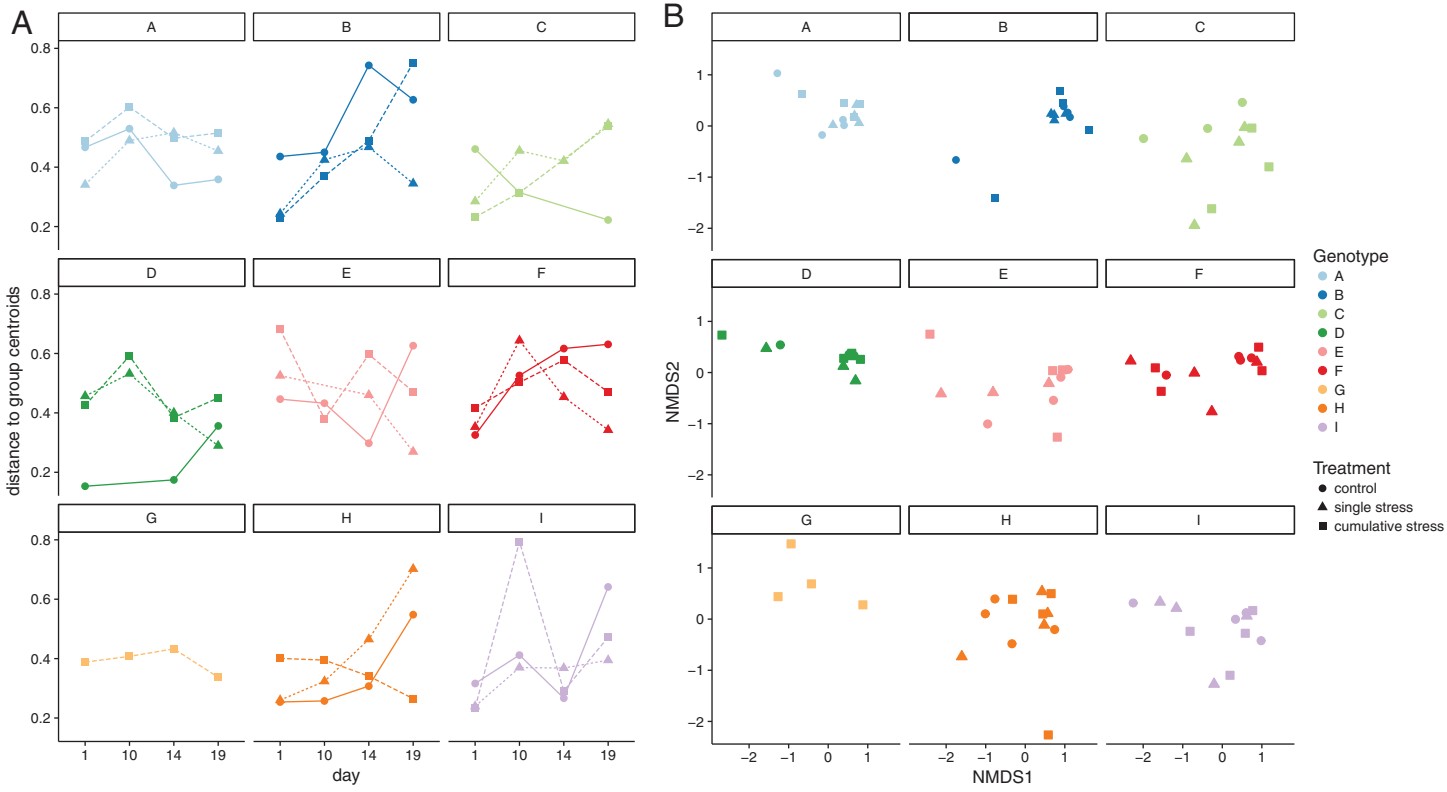

**Figure 3 Configurational and compositional stability of *Acropora tenuis* microbiome.** (A) Multivariate dispersion (heterogeneity) measured by the distance to the group centroid for each host-genotype (A–I) within each treatment (control, acute stress, and cumulative stress) over time (day 1, 10, 14, and 19). (B) Non-metric multidimensional scaling (NMDS) illustrating compositional similarity of sample replicates of each host-genotype (A–I) under different treatment conditions (control, single stress, and cumulative stress).

the cumulative stress treatment contained significantly lower protein and symbiont cell densities, while displaying significantly higher effective quantum yield compared to nubbins in the control and single stressor treatments (Tukey post hoc test; Table S2). Furthermore, effective quantum yield (one-way ANOVA, $F_{(8/91)} = 2.688$, $p = 0.0106$), symbiont cell densities (one-way ANOVA, $F_{(8/91)} = 4.334$, $p = 1.86 \times 10^{-4}$) and chlorophyll *a* concentrations (one-way ANOVA, $F_{(8/91)} = 2.773$, $p = 8.64 \times 10^{-3}$) varied significantly between host genotypes (Fig. 2B). Protein concentration, however, was unaffected by host genotype (one-way ANOVA, $F_{(8/91)} = 1.783$, $p = 0.0906$) and hence was the only holobiont health parameter solely affected by treatment.

## Microbial community response

The microbiome of *A. tenuis* remained highly stable across treatments, with no significant changes in the heterogeneity, also referred to as multivariate dispersion (one-way ANOVA, $F_{(2/93)} = 1.2107$, $p = 0.3026$; Fig. 3A), or in community composition (PERMANOVA, $p = 0.5156$, 10,000 permutations; Fig. 3B). However, the microbiome composition varied significantly between individual host genotypes (PERMANOVA, $p = 9.99 \times 10^{-5}$, 10,000 permutations), but was unaffected by treatment, sampling time point or tank effects when tested for each genotype individually (PERMANOVA
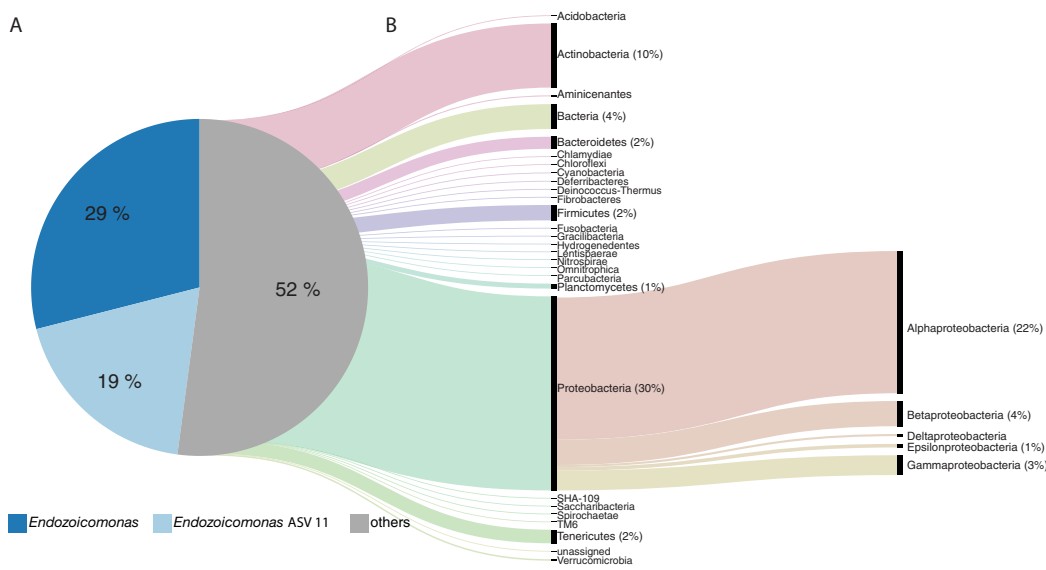

**Figure 4 The taxonomic composition of the *Acropora tenuis* microbiome.** (A) The *A. tenuis* microbiome was dominated by the bacterial genus *Endozoicomonas* (average relative abundance of 48%), with one *Endozoicomonas* ASV (ASV 11) present in all samples (average relative abundance of 19%). (B) The average contribution of the remaining microbiome (others) is displayed as an alluvial diagram, depicting the proportional contribution of bacterial phyla (classes for Proteobacteria). Mean relative abundances (%) are provided for bacterial taxa >1%.

with host-genotype as blocking factor, 10,000 permutations; Table S3). Similar results were obtained using presence/absence data (Fig. S1). Host genotype was the only significant factor, explaining 32.4% of the observed community variation (permutational ANOVA for db-RDA based on 1,000 permutations, $p = 9.99 \times 10^{-4}$; Fig. S2). Treatment and holobiont health parameters did not significantly contribute to the microbiome variation (Table S4). Furthermore, no significant correlation between similarity matrices based on microbiome composition and physiological holobiont health parameters was observed (chlorophyll *a*, protein, effective quantum yield, and symbiont cell density; Mantel statistic based on Pearson's product-moment correlation $r = -0.0238$, $p = 0.6243$, 10,000 permutations).

### *Endozoicomonas* assemblage

*Endozoicomonas* affiliated sequences comprised the majority of the *A. tenuis* microbiome, representing 48% (±29%) of the community (based on proportion of reads) and comprising 133 unique ASVs. One *Endozoicomonas* strain (ASV 11) was consistently present (100% of all samples) and highly abundant (19% ± 12%) throughout the experiment (Fig. 4). The *A. tenuis* microbiome also contained diverse bacteria affiliated with phyla including Proteobacteria (30%), Actinobacteria (10%), Firmicutes (2.4%), and Bacteroidetes (1.9%; Fig. 4).

The total relative abundance of *Endozoicomonas* was not affected by treatment (two-way ANOVA, $F_{(2/84)} = 0.473$, $p = 0.625$), sampling time point (two-way ANOVA, $F_{(3/84)} = 0.588$, $p = 0.625$) or the interaction of treatment-by-sampling time point

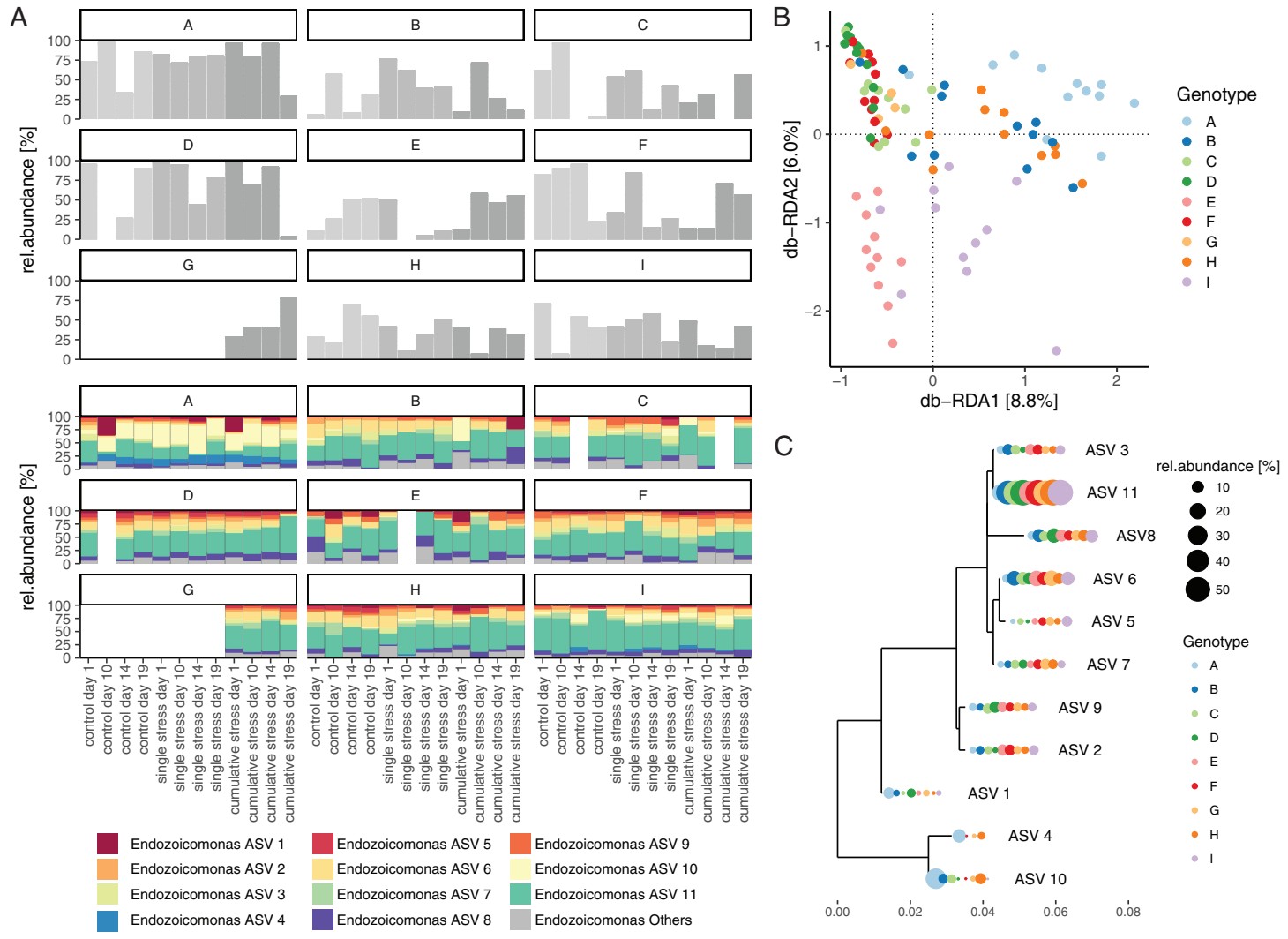

**Figure 5 Composition and distribution of *Endozoicomonas* assemblages.** (A) Total relative abundance of *Endozoicomonas* and the relative abundance distribution of the 11 most abundant *Endozoicomonas* ASVs associated with individual coral nubbins of each host genotype (A–I) under control, single stress, and cumulative stress conditions over time (day 1, 10, 14, and 19). (B) Distance-based Redundancy Analysis (db-RDA) quantifying the contribution of host-genotype to significantly explaining the observed compositional variation of the *Endozoicomonas* microbiome. (C) Phylogenetic tree of the 11 most abundant *Endozoicomonas* ASVs (including the ubiquitously present ASV 11) and their average relative abundance within a host genotype.

(two-way ANOVA, $F_{(6/84)} = 0.696$, $p = 0.654$). However, total relative *Endozoicomonas* abundance varied significantly between host genotypes (one-way ANOVA, $F_{(8/87)} = 3.741$, $p = 2.04 \times 10^{-4}$) and remained stable between treatments when tested for each genotype individually (within subject ANOVA, $F_{(2/85)} = 0.756$, $p = 0.473$; Fig. 5A).

The *Endozoicomonas* community composition also varied significantly between host genotypes (PERMANOVA, $p = 9.99 \times 10^{-5}$, 10,000 permutations; Fig. 5), however, was unaffected by treatment, sampling time point or tank (PERMANOVA with host-genotype as blocking factor, 10,000 permutations; Table S5). Furthermore, host-genotype significantly explained 26.4% of the observed compositional variability of

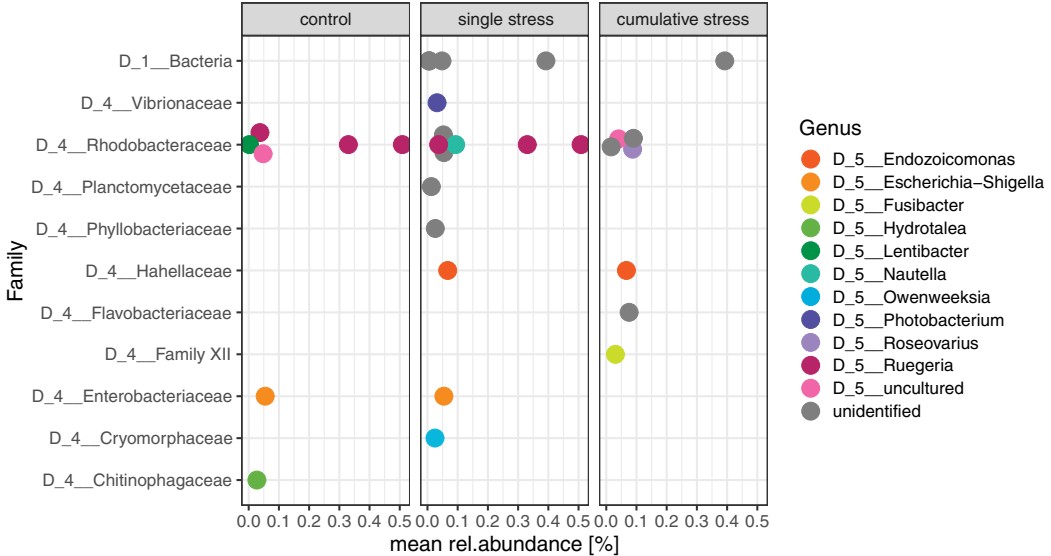

**Figure 6 Microbial indicators significantly associated with one and/or two treatments.** Indicators were identified based on their occurrence and abundance in coral tissue post stress exposure (excluding samples collected at day 1) using Indicator Value analysis. Each dot represents a single amplicon sequence variant (ASV), labeled with the taxonomic affiliation and their average relative abundance in the treatment group.

the *Endozoicomonas* community (permutational ANOVA for db-RDA based on 1,000 permutations, $p = 9.99 \times 10^{-5}$; Fig. 5B).

## Microbial indicators for environmental stress

Indicator value analysis was performed to assess if specific ASVs could be identified as indicators for environmental stress treatments. Despite the vast majority of ASVs (i.e., 4,598 ASVs) showing no response to experimental treatment, 26 ASVs were significantly associated ($p < 0.05$) with one and/or two treatment groups (Fig. 6; Table S6). The identified indicator ASVs belonged to the bacterial families Vibrionaceae, Rhodobacteraceae, Hahellaceae (genus *Endozoicomonas*), Planctomycetes, Phylobacteriaceae, Flavobacteriaceae, and Cryomorphaceae (Fig. 6).

## DISCUSSION

Elucidating the effect of host genotype on microbiome composition and understanding consequences of environmental change for holobiont stability is central to predicting the influence of host genetics on the stress tolerance of corals. Here, we followed the compositional stability of microbiomes associated with nine distinct *A. tenuis* genotypes when exposed to control, single and cumulative stress treatments over time. The *A. tenuis* microbiome varied significantly between coral genotypes, with host genotype being a much stronger driver of microbiome variation than environment. Similar host-genotype specificities have recently been described for sponge microbiomes (*Glasl et al., 2018*) and are also frequently reported for plant, crustacean, and human microbiomes (*Balint et al., 2013*; *Macke et al., 2017*; *Spor, Koren & Ley, 2011*). Traditional coral health parameters targeting the coral algal symbiont (i.e., chlorophyll *a* concentrations, symbiont

cell densities, effective quantum yield) were also significantly affected by host-genotype, although no correlation between these parameters and the microbiome was observed. This suggests that the *A. tenuis* microbiome composition remains largely unaffected by the performance and density of the algal symbiont, and that other host intrinsic factors (e.g., genetics) and/or the environmental life-history of individual genotypes fine-tune the microbiome composition.

*Endozoicomonas* form symbiotic partnerships with diverse marine invertebrates (*Neave et al., 2016a*). In corals, *Endozoicomonas* occur as dense clusters within the coral tissue and in some bacterial 16S rRNA gene profiling studies they can reach relative abundances as high as 95% of retrieved sequences (*Bayer et al., 2013*; *Neave et al., 2016a*; *Pogoreutz et al., 2018*). Loss of *Endozoicomonas* from the coral microbiome has been correlated with negative health outcomes for the coral host, though their direct effects on host fitness are unknown (*Bourne et al., 2008*; *Glasl, Herndl & Frade, 2016*; *Ziegler et al., 2016*). In *A. tenuis*, we detected no significant change in the relative frequency, alpha diversity, richness, and community composition of *Endozoicomonas* following exposure to non-lethal environmental stress. These results are consistent with findings for *Pocillopora verrucosa*, where *Endozoicomonas* remains the dominant symbiont even under bleaching conditions (*Pogoreutz et al., 2018*). In our study, the *Endozoicomonas* community generally exhibited high host-genotype specificity at the ASV level, though a single *Endozoicomonas* strain (ASV 11) was consistently shared among all coral nubbins and genotypes (including field control samples—data not shown). This ubiquitous strain likely represents a stable and consistent member of the resident *Endozoicomonas* community. A stable core is often described as a key feature of a symbiotic coral microbiome (*Ainsworth et al., 2015*; *Hernandez-Agreda, Gates & Ainsworth, 2017*), and despite being ubiquitously persistent between conspecific corals, the core characteristically only comprises a few members of the whole microbiome (*Hernandez-Agreda et al., 2018*).

While the *Endozoicomonas* community as a whole was not significantly affected by environmental treatment, one *Endozoicomonas* ASV was identified as a significant indicator for environmental stress. Similar environmental sensitivity has been reported for two prevalent *Endozoicomonas* species following exposure to elevated dissolved organic carbon (*Pogoreutz et al., 2018*). Although these *Endozoicomonas* affiliated ASVs show high sequence identity, small variations in the rRNA gene sequence can impact the biology and pathogenicity of bacteria (*Cilia, Lafay & Christen, 1996*; *Fukushima, Kakinuma & Kawaguchi, 2002*), hence single nucleotide variations (ASV level) may affect the functional role of microbes with flow on consequences for the coral holobiont. Shuffling and switching of *Endozoicomonas* strains may therefore provide the coral holobiont with an enhanced capacity to cope with shifting environmental conditions (*Neave et al., 2017*), although characterization of the symbiotic contribution made by *Endozoicomonas* to the coral host is required to better understand the ecological significance of these findings.

Recent studies have highlighted the potential for coral microbiomes to act as sensitive markers for environmental disturbance (*Glasl, Webster & Bourne, 2017*;

*Roitman, Joseph Pollock & Medina, 2018*). Here, we showed that a small number of ASVs, including taxa commonly reported to increase under host stress (i.e., Vibrionaceae, Rhodobacteraceae (*Ben-Haim et al., 2003*; *Bourne, Morrow & Webster, 2016*; *Sunagawa, Woodley & Medina, 2010*)), were significantly associated with the tissue of *A. tenuis* exposed to single and cumulative stress treatments. However, despite the potential diagnostic value of these ASVs, host genotype overwhelmed any overarching effect of environment on the coral microbiome. This high divergence in the microbiome between conspecific corals is likely to hinder our ability to detect fine-scale variation of sensitive microbial indicator taxa. Therefore, unless host-genotype independent microbial indicators can be identified and validated, the efficacy of integrating microbial community data into coral health monitoring initiatives appears unfeasible due to high compositional variability between microbiomes of conspecific corals.

## CONCLUSIONS

This study shows that the *A. tenuis* microbiome varies significantly between host individuals (genotypes) and that these genotype-specific communities persist during exposure to non-lethal environmental disturbances. Consideration of microbiome-by-host genotype-by-environment effects is therefore needed to elucidate how intraspecific variations of the microbiome affect the susceptibility of corals to environmental stress and disease. Furthermore, microbial variability between individual coral genotypes may cloud our ability to identify universal microbial changes during periods of adverse environmental conditions. This is particularly relevant if establishing sensitive microbial indicators for sub-lethal environmental disturbances (tested in this study), since the observed stability of the coral microbiome combined with the host genotype specificity likely precludes the robust assignment of microbial indicators across broad scales.

## ACKNOWLEDGEMENTS

We thank Victoria Lydick and Andrew Ball for their help during the experiment. We also acknowledge the technical support provided by the National SeaSimulator, and are particularly grateful to Craig Humphrey and Andrea Severati for their support with the experimental system. We thank Pedro R. Frade for inspiring scientific discussions on the coral microbiome and photobiology of corals and Jonathan Barton for his insights into coral husbandry.

### Funding

The study was funded by the Advance Queensland PhD scholarship, AIMS@JCU PhD scholarship and the GBRMPA Science for Management Award awarded to Bettina Glasl. The funders had no role in study design, data collection and analysis, decision to publish, or preparation of the manuscript.

## Grant Disclosures
The following grant information was disclosed by the authors:
Advance Queensland PhD scholarship.
AIMS@JCU PhD scholarship.
GBRMPA Management Award awarded to Bettina Glasl.

## Competing Interests
The authors declare that they have no competing interests.

## Author Contributions
- Bettina Glasl conceived and designed the experiments, performed the experiments, analyzed the data, contributed reagents/materials/analysis tools, prepared figures and/or tables, authored or reviewed drafts of the paper, approved the final draft.
- Caitlin E. Smith conceived and designed the experiments, performed the experiments, approved the final draft.
- David G. Bourne conceived and designed the experiments, authored or reviewed drafts of the paper, approved the final draft.
- Nicole S. Webster conceived and designed the experiments, authored or reviewed drafts of the paper, approved the final draft.

## Field Study Permissions
The following information was supplied relating to field study approvals (i.e., approving body and any reference numbers):

Corals were collected under the permit G12/35236.1 granted by the Great Barrier Reef Marine Park Authority to the Australian Institute of Marine Science.

## Data Availability
Demultiplexed sequences and metadata are available from the Sequence Read Archives (SRA) under accession number PRJNA492377.

Furthermore, ASV abundance table, metadata and ASV taxonomies are available as Supplemental Files.

R code is available from the GitHub repository under https://github.com/bglasl/Git_4_Coral

## Supplemental Information
Supplemental information for this article can be found online at http://dx.doi.org/10.7717/peerj.6377#supplemental-information.

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
