# Peer review of "Disentangling the effect of host-genotype and environment on the microbiome of the coral Acropora tenuis"

_PeerJ, doi:10.7717/peerj.6377_

## Round 0.1 · original submission · Major Revisions

Your manuscript has been evaluated by two expert reviewers and their comments can be seen below. Both reviewers have very favourable comments on the manuscript, however they also have a number of usefeul suggestions and relevant observations, all of which you should consider in a revision.

Reviewer 1 ·

Basic reporting

The study is interesting and a pleasant read. English is used throughout, the language is professional. The article structure is professional. The introduction is well referenced, and properly and evenly cited. The description of the figures is adequate. The BioProject on NCBI at which the raw seq data are deposited is accessible. The manuscript is self-contained and represents an appropriate unit of publication.

There reporting of basic sequencing results and presentation of bacterial community composition will benefit from adding more detail.

Major comments:
1. I am missing basic reporting of sequencing data, such as total numbers of seqs or reads before and after QC/removal of chimeric seqs and other unwanted seqs (like mitochondria, eukaryotes, lab contaminants), average amplicon length, etc. In addition, it would be informative both from the viewpoint of a reviewer and a reader to have an ASV abundance table (like an OTU abundance table) provided. This could e.g. be a .csv or .txt table provided in the supplement.
2. Dealing with kit contaminants, and reporting thereof. Lines 180-181: The Qiagen kits are often contaminated with bacterial DNA/amplicons, which may affect/distort bacterial community datasets (cf. Salter et al. 2014 BMC Biology). The best way to deal with this is to include ‘negative control’ mock samples (DNA extractions without template, using only kit reagents) and mock PCRs (PCR products without template, using only master mix reagents) and sequence them along with your samples. I strongly encourage the authors to add brief details in whether and how they’ve dealt with potential lab/kit contamination. I am a bit concerned about the proportion of Actinobacteria shown in the alluvial diagram (Fig. 4).
3. Lines 266 and following: Endozoicomonas may be the most abundant bacterial genus present in your sequence data, but associated seqs make up only about half of the microbiome (48 % on average). It would be informative to add some numbers for mean relative abundances of other bacterial groups (on the class and family or genus level) in the manuscript text, starting from line 272. Please also describe in the results section how many different Endozoicomonas ASVs were identified in the analysis, and what are their relative abundances (information can be drawn from figures, but it is a bit frustrating for the reader to not have it briefly compiled in the ms text).
4. in this context, it would be informative to provide a stack bar plot showing the microbiome composition (e.g., rel. abundances of xx most abundant ASVs + ‘others’) on the ASV or family level across coral genotypes in the manuscript to visualize the genotype-specific patterns.
5. It is good practice to provide basic information (e.g., a table) showing diversity and richness indices of total bacterial communities over samples or treatments.
6. Line 347: ‘some bacteria were significantly more abundant in treatments’ – the results state that the microbiome composition did not differ between treatments. Please provide information for significantly differentially abundant data in results section if you tested individual variants/families etc. If this is referring to indicator taxa, please correct statement – indicator taxa are significantly associated with a condition, environment, etc. This is not necessarily about abundance (rare members can be indicators), but persistent association.

Minor comments:
1.Lines 182-183: It would be helpful to provide more detail on PCR conditions along with sequences for F and R primer including illumina overhangs rather than only provide a reference.
2. Lines 201-202: Please mention that the bioproject uploaded in the SRA is available on NCBI.
3. The authors keep switching between the terms ‘effective quantum yield’ and ‘photochemical efficiency’. I strongly encourage the authors to consistently use one term only (Quantum yield – effective or maximum, whichever applies – see comment below in experimental design) throughout the manuscript so as not to confuse readers who are not familiar with this kind of measurement.
4. Lines 290-293: It would be great to add a sentence here on which of these bacterial groups are significantly associated with which experimental conditions. It is a bit frustrating for the reader to have to go get it from the supplement.
5. Line 325: the authors argue that Endozoicomonas ASV1 might be a stable and persistent member of the A. tenuis they are working on. The authors can easily get information on the core microbiome out of their data. They could even run the analysis together with sequences from field control corals (which seem to be available) and add this detail to the manuscript. This should not be considered as a requirement to have the manuscript accepted, but since this is a calculation that doesn’t take up much time and is discussed by the authors anyways, they might as well include core microbiome data.
6. Figure 6: the color gradient used for the indicator taxa in the plot is a bit hard to distinguish, in particular the greens and the blues. I encourage the authors to modify the color scheme to help the reader.

Experimental design

The research presented here is within the Aims and Scope of PeerJ.

The research goal is defined (effect of host genotype-by-environment interactions on the microbiome of Acropora tenuis), relevant, and meaningful. It clearly addresses an existing knowledge gap (combined influence of host-genotype and environmental stress on the microbial community composition of corals). The experiment has been developed and executed with rigor, to a high technical standard. I cannot identify any obvious flaws. The methods described contain sufficient detail and information to replicate the experiment.

There are a couple points I would like to get clarification on:

1. PAM measurements. The authors state they have measured effective quantum yield Fv/Fm from light adapted corals. As described by Chakravati et al. 2017 GCB, Fv/Fm is the maximum, i.e. dark-adapted quantum yield – the light-adapted or effective quantum yield of PSII is delta Fv/F’m. Please correct the measures accordingly to what it is you measured.
2. coral surface area quantification - Line 150-151: the authors used a single wax coating. Previous studies have shown that two wax coatings (as described in Naumann et al. 2009) are more precise. Please explain why you decided to use a single wax coating instead.

Validity of the findings

Replication is meaningful. Data is statistically sound and controlled, but I have one comment (see below)

Absence of bacterial community responses (‘stable’ or ‘inflexible’ bacterial microbiomes) under stress have previously been observed in different corals under different stress scenarios. The authors found that bacterial communities however differ significantly between coral genotypes. This is not necessarily surprising – as the authors correctly state in the introduction, genotype-specific physiological responses have been known for corals for a good long time. There is one point that I would like the author to consider to test (if they haven't done so yet).

1.Lines 251 and following: ‘stable’ bacterial communities. Have you tested for differences in the bacterial community after omitting the most abundant bacteria (Endozoicomonas ASV1, plus potentially others) from analysis to see whether any changes occur in bacteria with lower abundance? Rare microbes might matter – and sometimes less abundant bacterial taxa can exhibit changes under stress while the most abundant taxa remain ‘stable’, but this response can sometimes be statistically masked by high abundances of dominant taxa.

conclusions are well stated and linked to the original research goal.

Additional comments

Overall, the manuscript is well written. The data support the conclusion presented by the authors (but I encourage the authors to check if bacterial communities show patterns once Endos are omitted from analysis). The language of the manuscript is professional and standard English. There are some sections in methods and results where it is not entirely clear whether measurements encompass the holobiont or symbiont only, and whether statistically significant effects where identified between all time points or not, which I have detailed below.

This study fits well in the scope of PeerJ.

Abstract:
‘Planctomytaceae’ – do the authors mean ‘Planctomycetes’ instead?

Line 120: wouldn’t it be acclimation rather than acclimatization if the corals have to get used to new conditions in a new environment (I understand they were moved to a different environment, the experimental tank?)

Line 145: add ‘amplicon’ to 16S rRNA gene sequencing

Line 154: ‘Host physiological measurements’ is not a good way to describe the following section, as three of the four measurements (photochemical efficiency, zoox cell counts, and chl a content) are specific measurements of the Symbiodiniaceae and the fourth (total protein content) is on the holobiont level. If zoox and host cells/matter would have been separated into fraction and the host fraction would have been assessed separately, that would be a measure of host physiology. I suggest to rephrase (e.g., Physiology of Symbiodiniaceae and the holobiont). This also applies to the use of ‘host health parameters’ in lines 215 and 259 (and a couple of other locations in the manuscript).

Lines 236 and following: please detail between which time points the significant effects occurred. If there were statistically significant differences between each time point, please state this clearly.

Lines 351-354: ‘Therefore, unless host-genotype independent microbial indicators can be identified and validated, the efficacy of integrating microbial community data into coral health monitoring initiatives appears unfeasible…’ – I strongly agree with the authors' assessment, and I appreciate them being upfront.

Reviewer 2 ·

Basic reporting

The manuscript is well written, clear and well supported by the cited literature. However, justification of the various environmental stressors and the comparison of a single versus cumulative stress are not introduced in the introduction. Please provide some justification. These can be brief, with one or two references providing evidence that each is a relevant stressor. Also, a brief introduction to single versus cumulative stress should be included. The raw data is available in the SRA of NCBI, and the figures and tables are well set out. There are several minor issues with fine editing:
Abstract
- line 45 This sentence is unclear. Instead something like: The A. tenuis microbiome was dominated by Endozoicomas with, on average, 48% of ASVs assigned to this genus, with ASVs significantly different between host individuals.
Introduction
- line 46 Should this be coral hosts?
Introduction
- line 89/90 the semi colon should be after the references
- justification/reasoning behind choice of stressors is missing
Methods/Results
- be consistent with spacing between numbers and units (e.g. 24h should be 24 h, and in other places with other units).

Experimental design

- The experimental design is robust, however the authors have not accounted for tank effects. These can skew the interpretation of tank experiments, and the authors should address this either by testing for tank effects or by explaining why these were not accounted for in the statistical analysis.
- The description of the quantification of various measures from the airbrushed tissue: Was a consistent volume of PBS used? If not, could the cell counts be influenced by the volume used here? Also, Was the tissue allocated equally among the various subsequent analyses? If not, it would also influence the results of some of these measures.
- Line 198 were eukaryote and unknown sequences also removed?

Validity of the findings

- Lines 207 and following: please clarify the ANOVA factors, indicating where 1-way, 2-way or other models were applied, and give details of which factors are being tested. The design appears to be a 3 factor design with time, treatment and genotype as the 3 factors. However, the authors have elected to apply a variety of different types of ANOVAs and PERMANOVAs, which should be clearly justified throughout. In all cases, the interactions should also be reported. The main ANOVA and PERMANOVA tables should be included in the supplementary material, as well as the post-hoc and pairwise tests.
- The authors appear to have mastered the various analyses that have been applied to the data, however lack of clarity in the manipulation of the 3 factors within the experiment and the subsequent statistical analyses (outlined above) result in some uncertainty around the outputs and the interpretation of the results. This includes any tank effects that may be influencing the outcomes.
- Conclusions are well stated and linked to results, but the results need tightening up, as outlined above.

Additional comments

A nicely designed study with several interesting results. This paper is likely to be of broad interest to the field of coral/microbial ecology.

---

## Round 0.2 · Minor Revisions

Thank you for resubmitting your manuscript. In my opinion you have done an excellent job of incorporating the reviewers´ observations and suggestions and am pleased with the revised manuscript. My only observation is that Symbiodiniaceae is a Family name and therefore should not be italicized; only genus and species are italicized. This is a quick fix and once it has been taken care of, I will suggest that the manuscript be accepted for publication.

---

## Round 0.3 · accepted · Accept

I am satisfied with the changes made to the manuscript.

#